

# Association of sleep duration and sleep quality with hypertension in oil workers in Xinjiang

Fen Yang[1], Yuanyue Zhang[1], Ruiying Qiu[1] and Ning Tao[1,2]

[1] School of Public Health, Xinjiang Medical University, Xinjiang, China
[2] Clinical Postdoctoral Mobile Station, Xinjiang Medical University, Xinjiang, China

## ABSTRACT

**Objective**. The aim of this study is to explore sleep status and hypertension among oil workers in Xinjiang, China. It may provide new ideas and basis for the precise prevention and treatment of hypertension in occupational population.

**Methods**. Sleep status and hypertension were investigated in 3,040 workers by a multi-stage cluster sampling method in six oil field bases in Karamay City, Xinjiang. The Pittsburgh Sleep Quality Index was used to evaluate the sleep status of workers. Logistic regression was used to analyze the relationship between sleep duration and sleep quality, and hypertension. Stratified analysis was also performed.

**Results**. Our results show: 1. Insufficient sleep duration ($OR = 1.51$, 95% CI [1.19–1.90]) and poor sleep quality ($OR = 1.78$, 95% CI [1.33–2.38] were positively associated with hypertension. 2. Stratified analysis indicated insufficient sleep duration was associated with increased risk of hypertension in females ($OR = 1.54$, 95% CI [1.16–2.04]) than males ($OR = 1.49$, 95% CI [1.00–2.23]), and the risk of hypertension in the group <30 years old ($OR = 9.03$, 95% CI [2.32–35.15]) was higher than that in the group of 30–45 years old ($OR = 1.59$, 95% CI [1.14–2.20]). However, in the group > 45 years old, sleeping > 8 h was associated with increased risk of hypertension ($OR = 3.36$, 95% CI [1.42–7.91]). Oil workers doing shift work had a higher risk of hypertension ($OR = 1.55$, 95% CI [1.16–2.07]) to no shift work ($OR = 1.48$, 95% CI [1.02–2.15]). The risk of hypertension in the group with < 10 years of service ($OR = 4.08$, 95% CI [1.92–8.83]) was higher than that in the group with length of service of 10–20 years ($OR = 2.79$, 95% CI [1.59–4.86]). Poor sleep quality was associated with risk for hypertension in females ($OR = 1.78$, 95% CI [1.26–2.49]), those doing shift work ($OR = 1.70$, 95% CI [1.17–2.47]), those with length of service of > 20 years ($OR = 1.64$, 95% CI [1.18–2.27]). The risk of hypertension in the group 30–45 years old is higher than that in the group > 45 years old ($OR_{30–45 \text{ years old}} = 1.71$, 95% CI [1.10–2.66]; $OR > 45 \text{ years old} = 1.60$, 95% CI [1.09–2.34]).

**Conclusion**. Insufficient sleep duration and poor sleep quality are the potential factors affecting hypertension in Xinjiang oil workers.

Corresponding author
Ning Tao, 38518412@qq.com

## INTRODUCTION

Hypertension is a non-communicable disease and the most important risk factor leading to death from cardiovascular disease. At present, although the etiology of hypertension is unclear, factors identified as influencing hypertension include age, gender, race, lack of exercise, obesity, sodium intake, alcohol, and occupational stress (*Bergmann, Gyntelberg & Faber, 2014*). In recent years, some studies have found that sleep is also closely related to the occurrence of hypertension. Sleep has important effects on the cardiovascular system function (*Wang et al., 2017*), as well as on physiological pathology (*Christina & Fernandez-Mendoza, 2018*). However, the rapid development of human society and changes in lifestyle have led to a decrease in average sleep duration and sleep quality (*Zheng, Chen & Chen, 2014*). In line with the National Health Interview Survey, short sleep duration is associated with an increased risk of hypertension among American adults, and this relationship is dependent on age and BMI (*Oluwatimilehin et al., 2019*). *Wu et al. (2019)* conducted a recent survey on hypertension in daytime and nighttime. In that study, the results showed that only males' sleep duration was associated with risk of hypertension. On the other hand, studies by *Wu et al. (2016)* found that lack of sleep associated with hypertension is prevalent only in females. These contrary results suggest there is a complex and controversial relationship between sleep and hypertension. Oil workers are a specific occupational group often involving shift work. Sleep problems often occur when there is a serious imbalance between the natural circadian rhythm and the shift system. Previous studies have shown that not only is shift work associated with adverse health effects by disturbing circadian rhythm, but may also impact sleep quality and be linked with diabetes and hypertension (*Choi et al., 2019*). *Wang & Bai (2014)* conducted statistical analysis of the health status of oil workers in Ningxia, China, finding the prevalence of hypertension of 21.43%. Such a high incidence suggests improving the sleep status and controlling for the risk of hypertension among desert oil workers are matters of urgency, owing to the specific characteristics of this occupational group. Oil fields are often located in remote areas, including in deserts and off shore. As such, they are often affected by challenging weather conditions. Moreover, oil rigs typically operate 24 h a day, thereby requiring shift work with oil field workers often working one week on and one week off.

Therefore, this cross-sectional study set out to investigate the sleep status of Chinese oil workers with the aim of exploring the association between sleep duration and sleep quality on hypertension. Confounding factors were analyzed by a logistic regression model. Sleep duration and sleep quality, and their association with hypertension were further investigated by stratified analysis, which provided a more comprehensive theoretical basis for identifying preventive measures of hypertension among oil workers in Xinjiang.

## MATERIALS & METHODS

### Study setting and participants

According to $N = \mu_\alpha^2 \rho (1-\rho)/\delta^2$ [$\rho$ is the prevalence of hypertension (according to the research results of Lu et al., the value is 37%), $\alpha = 0.05$ (two-sided), $\delta = 0.018$] that $N = 2763$. Taking into account the censorship of 10%–20%, 3,100 oil workers were finally

selected. Cluster sampling was used to select the study population. A cross-sectional survey of 3,100 oil workers in six oil field bases in Karamay, Xinjiang, China, was conducted. One district (Karamay District) was randomly selected from four districts (Karamay District, Baijiantan District, Wuerhe District, and Dushanzi District) under the jurisdiction of Karamay City. Sixteen oil fields in this area were numbered 01 ~16, according to the random number table method. Six oil fields corresponding to the first column of row 27 in the random number table were selected. All participants provided written informed consent to a questionnaire survey. Excluding 33 with incomplete questionnaires and 27 oilfield workers who failed to measure blood pressure by professional doctors in Karamay Central Hospital, 3,040 participants were finally included. The inclusion criteria were oil field workers aged between 20 and 60 with work experience of at least one year who agreed to participate in this survey. The exclusion criteria were those with severe organic diseases (that occurs in a certain organ or a certain tissue system of the body caused by a variety of reasons, which causes permanent damage to the organ or tissue system), mental illness, and genetic diseases. This study was approved by the Ethics Review Committee of the First Affiliated Hospital of Xinjiang Medical University (Ethics number: 2015006).

## Study methods
### Socio-demographic characteristics
Each subject participated in a structured questionnaire survey, which included factors such as gender (female, male), age (< 30, 30–45, > 45 years old), ethnicity (Han, others), marital status (unmarried, married, divorced/widow/widower), income (< 5000 RMB, ≥ 5000 RMB), educational level (above senior high school, below technical school), job status (junior level, intermediate level, senior level), length of service (< 10, 10–20, > 20 years), smoking status (non-smoker, smoker: smoking ≥ 1 cigarette per day for six months or more), alcohol consumption (non-drinker, drinker: drinking ≥ 2 times a week with alcohol intake ≥ 50 g per drinking session regularly ≥ 1 year), BMI (< 18.5, 18.5–24, >24), and shift work status (no shift work, shift work: regular working hours other than 10:00-19:00 and lasting for more than 1 year).

### Sleep status
The Pittsburgh Sleep Quality Index (PSQI) was used to assess participants' subjective sleep quality and sleep duration (*Buysse et al., 1989*). The survey used the Chinese version of the Pittsburgh Sleep Quality Index (CPSQI), with a Cronbach's alpha of 0.89 and a KMO of 0.91 (*Tsai et al., 2005*). The CPSQI comprises seven factors and 19 items. Each item is scored from 0 to 3 according to a 4-level Likert scale. In this study, a CPSQI ≤ 5 was considered good sleep quality, whereas a score of > 5 was regarded as poor sleep quality. Sleep duration was classified into "insufficient" (< 7 h), "normal" (7–8 h), and "long" (> 8 h) based on previous reports (*Tsai et al., 2005*).

### Ascertainment of hypertension
Blood pressure was measured by a professional doctor in the Physical Examination Department of Karamay Central Hospital. The blood pressure and diagnosis of hypertension in each participant were determined after measuring resting blood pressure

three times during different times of the same day. Individuals were defined as having hypertension if they met one of the following standards (*Chinese Journal of Cardiovascular Medicine, 2018*): (1) systolic blood pressure $\geq$ 140 mm Hg and/or diastolic blood pressure $\geq$ 90 mm Hg; (2) self-reported hypertension diagnosed by a physician and current antihypertensive treatment during the previous two weeks.

### Statistical analysis

The raw data were organized in Excel sheets. All the statistical analyses of this research were performed using SPSS version 24.0. The categorical data are tabulated with frequencies and percentages, and a Pearson chi-square test was computed to compare differences between groups. Logistic regressions were performed to evaluate the relationship between sleep duration and sleep quality with hypertension. The level of significance was set at 0.05.

## RESULTS

### Basic characteristics of the sample population

The data of all 3,040 participants in this study are presented in Table 1. The overall prevalence of hypertension was 15.33% among all participants. Participants reporting sleep of $< 7$ h accounted for 26.51%, of which 21.00% reported hypertension. Participants reporting poor sleep quality accounted for 78.28%, of which 16.50% reported hypertension. Hypertension showed significant association with all characteristics except for income level in the chi-squared test ($P < 0.001$).

### Sleep duration, sleep quality, and hypertension

After adjusting for confounding factors including gender, age, ethnicity, educational level, job status, marital status, income, smoking status, alcohol consumption, shift work status, and BMI, insufficient sleep duration and poor sleep quality showed a statistically significant association with hypertension ($OR_{sleep\ duration}$ =1.51, 95% CI [1.19–1.90]; $OR_{sleep\ quality}$ = 1.78, 95% CI [1.33–2.38]). The results indicated that age ($OR_{30-45\ years\ old}$ = 2.00, 95% CI [1.03–3.91]; $OR_{>45\ years\ old}$ = 2.91, 95% CI [1.43–5.91]), gender ($OR$ = 2.08, 95% CI [1.53–2.81]), shift work status ($OR$ = 2.22, 95% CI [1.78–2.78]), and length of service ($OR_{10-20\ years}$ = 2.34, 95% CI [1.43–3.81]; $OR_{>20\ years}$ = 3.87, 95% CI [2.40–6.24]) were risk factors associated with hypertension ($P < 0.001$). Income ($OR$ = 0.65, 95% CI [0.52–0.81]) and ethnicity ($OR$ = 0.61, 95% CI [0.50–0.79]) were protective factors for hypertension ($P < 0.01$). The risk of hypertension was higher in females than in males ($P < 0.001$). The educational level, job status, marital status, smoking status, alcohol consumption, and BMI did not show significant impact associated with risk of hypertension ($P > 0.05$). According to Hosmer-Lemeshow goodness-of-fit test, $\chi^2$ =14.929, $P = 0.061$ $> 0.05$, suggesting that the model has a good degree of fit and calibration ability; The area under the curve of the model AUC is 0.782 $> 0.75$, 95% CI [0.758–0.801], suggesting that the prediction model has a good distinguishing ability (Table 2).

### Stratified analysis

The logistic regression analysis indicated that age, gender, shift work status, and length of service were risk factors for hypertension, so we stratified according to these factors and

**Table 1** The basic characteristics of subjects with or without hypertension.

| Item | Frequency | Hypertension | | $\chi^2$ | P value |
|---|---|---|---|---|---|
| | | Yes | No | | |
| Sleep duration, h | | | | | |
| 7–8 | 2147 | 283 (13.2) | 1864 (86.8) | 26.76 | <0.001 |
| <7 | 806 | 168 (21.0) | 638 (79.0) | | |
| >8 | 87 | 15 (15.3) | 72 (84.7) | | |
| Sleep quality | | | | | |
| Poor | 2380 | 392 (16.5) | 1988 (83.5) | 11.01 | 0.001 |
| Good | 660 | 74 (11.2) | 586 (88.8) | | |
| Gender | | | | | |
| Male | 1325 | 134 (10.1) | 1191 (89.9) | 49.23 | <0.001 |
| Female | 1715 | 332 (19.4) | 1383 (80.6) | | |
| Age, years | | | | | |
| ≤30 | 452 | 14 (3.4) | 438 (96.6) | 119.79 | <0.001 |
| 30–45 | 1625 | 216 (12.9) | 1409 (87.1) | | |
| >45 | 963 | 236 (24.5) | 727 (75.5) | | |
| Ethnicity | | | | | |
| Han | 2083 | 350 (16.8) | 1733 (83.2) | 11.07 | 0.001 |
| Others | 957 | 116 (12.1) | 841 (87.9) | | |
| Education level | | | | | |
| Above senior high school | 1066 | 231 (21.7) | 835 (78.3) | 50.86 | <0.001 |
| Below technical school | 1974 | 235 (11.9) | 1739 (88.1) | | |
| Job status | | | | | |
| Junior level | 865 | 110 (12.7) | 755 (87.3) | 34.27 | <0.001 |
| Intermediate level | 679 | 70 (10.3) | 609 (89.7) | | |
| Senior level | 1496 | 286 (9.4) | 1210 (80.9) | | |
| Marital status | | | | | |
| Unmarried | 268 | 20 (7.5) | 248 (92.5) | 15.27 | <0.001 |
| Married | 2483 | 393 (15.8) | 2090 (84.2) | | |
| Divorced/Widow/Widower | 289 | 53 (18.3) | 236 (81.7) | | |
| Income, RMB | | | | | |
| <5000 | 1580 | 223 (14.1) | 1357 (85.9) | 3.74 | 0.053 |
| ≥5000 | 1460 | 243 (16.6) | 1217 (83.4) | | |
| Smoking status | | | | | |
| Smoker | 1845 | 243 (13.2) | 1602 (86.8) | 16.84 | <0.001 |
| Non-smoker | 1195 | 223 (18.7) | 972 (81.3) | | |
| Alcohol consumption | | | | | |
| Drinker | 1785 | 319 (17.9) | 1466 (82.1) | 21.53 | <0.001 |
| Non-drinker | 1255 | 147 (11.7) | 1108 (88.3) | | |
| Shift work status | | | | | |
| Shift work | 1503 | 283 (18.8) | 1220 (81.2) | 28.06 | <0.001 |
| No shift work | 1537 | 183 (11.9) | 1354 (88.1) | | |

**Table 1** (*continued*)

| Item | Frequency | Hypertension | | $\chi^2$ | *P value* |
|---|---|---|---|---|---|
| | | Yes | No | | |
| BMI, kg m$^{-2}$(Body mass index, kg m$^{-2}$) | | | | | |
| <18.5 | 84 | 3 (3.6) | 81 (96.4) | 104.45 | <0.001 |
| 18.5–24 | 1604 | 156 (9.7) | 1448 (90.3) | | |
| >24 | 1352 | 307 (22.7) | 1045 (77.3) | | |
| Length of service, years | | | | | |
| <10 | 850 | 35 (4.1) | 815 (95.9) | 141.32 | <0.001 |
| 10–20 | 566 | 73 (12.9) | 493 (87.1) | | |
| >20 | 1624 | 358 (22.0) | 1266 (78.0) | | |

analyzed the effect of sleep duration and sleep quality on hypertension. Insufficient sleep duration was found to be associated with increased risk of hypertension in females ($OR = 1.54$, 95% CI [1.16–2.04]) higher than males ($OR = 1.49$, 95% CI [1.00–2.23]), with the risk of hypertension in the group < 30 years old higher than that in the group 30–45 years old ($OR_{<30 \text{ years old}} = 9.03$, 95% CI [2.32–35.15]; $OR_{30-45 \text{ years old}} = 1.59$, 95% CI [1.14–2.20]). However, in the group > 45 years old ($OR = 3.36$, 95% CI [1.42–7.91]), sleeping for > 8 h was associated with increased risk of hypertension, with the risk of hypertension in the group with < 10 years of service being higher than that in the group with the length of service of 10–20 years ($OR_{<10 \text{ years}} = 4.08$, 95% CI [1.92–8.83]; $OR_{10-20 \text{ years}} = 2.29$, 95% CI [1.59–4.86]). Oil workers doing shift work had risk of hypertension ($OR = 1.55$, 95% CI [1.16–2.07]) to no shift work ($OR = 1.48$, 95% CI [1.02–2.15]). Compared with a good sleep quality, poor sleep quality was shown to be associated with increased risk of hypertension in females ($OR = 1.78$, 95% CI [1.26–2.49]), length of service > 20 ($OR = 1.64$, 95% CI [1.18–2.27]), and shift work ($OR = 1.70$, 95% CI [1.17–2.47]); with the risk of hypertension in the group 30–45 years old higher than that in the group > 45 years old ($OR_{30-45 \text{ years old}} = 1.71$, 95% CI [1.10–2.66]; $OR_{>45 \text{ years old}} = 1.60$, 95% CI [1.09–2.34] (Table 3)).

## DISCUSSION

In this cross-sectional study, the associations of sleep duration and sleep quality with the prevalence of hypertension among oil workers in Xinjiang were investigated. We adjusted for gender, age, length of service, shift work status and other confounding factors, and performed logistic regression analysis to assess the relationship between sleep duration and sleep quality, and hypertension. We found that insufficient sleep duration and poor sleep quality are associated with hypertension in Xinjiang oil workers. Moreover, their associations also analyzed by stratification, our results showed that this phenomenon was found in different gender, age, length of service and shift status.

### Relationship between hypertension sleep and hypertension in oil workers

Hypertension currently affects 26.4% of adults worldwide, and it is a leading risk factor for mortality (*Huang et al., 2012*). Published in *the Lancet* in 2017, *Lu et al. (2017)* conducted

**Table 2** Results of logistic regression analysis investigating the association between sleep duration, sleep quality and the risk of hypertension among oil workers.

| Item | b | SE | Wald $\chi^2$ | P value | OR (95%CI) |
|---|---|---|---|---|---|
| Sleep duration, h | | | | | |
| 7–8 | – | – | – | – | 1.00 |
| <7 | 0.409 | 0.119 | 11.785 | 0.001 | 1.51(1.19–1.90) |
| >8 | 0.424 | 0.318 | 1.772 | 0.183 | 1.53(0.82–2.85) |
| Sleep quality | | | | | |
| Good | – | – | – | – | 1.00 |
| Poor | 0.576 | 0.148 | 15.166 | <0.001 | 1.78(1.33–2.38) |
| Gender | | | | | |
| Male | – | – | – | – | 1.00 |
| Female | 0.730 | 0.154 | 22.389 | <0.001 | 2.08(1.53–2.81) |
| Age, years | | | | | |
| <30 | – | – | – | – | 1.00 |
| 30–45 | 0.694 | 0.342 | 4.125 | 0.042 | 2.00(1.03–3.91) |
| >45 | 1.066 | 0.362 | 8.675 | 0.003 | 2.91(1.43–5.91) |
| Ethnicity | | | | | |
| Han | – | – | – | – | 1.00 |
| Others | −0.498 | 0.132 | 14.227 | <0.001 | 0.61(0.50–0.79) |
| Educational level | | | | | |
| Above senior high school | – | – | – | – | 1.00 |
| Below technical school | −0.199 | 0.121 | 2.684 | 0.101 | 0.82(0.65–1.04) |
| Job status | | | | | |
| Junior level | – | – | – | – | 1.00 |
| Intermediate level | −0.300 | 0.158 | 3.602 | 0.058 | 0.74(0.54–1.01) |
| Senior level | −0.050 | 0.135 | 0.138 | 0.710 | 0.95(0.73–1.24) |
| Marital status | | | | | |
| Unmarried | – | – | – | – | 1.00 |
| Married | −0.305 | 0.293 | 1.086 | 0.297 | 0.74(0.42–1.31) |
| Divorced/Widow/Widower | −0.095 | 0.333 | 0.082 | 0.755 | 0.91(0.47–1.75) |
| Income, RMB | | | | | |
| <5000 | – | – | – | – | 1.00 |
| ≥5000 | −0.427 | 0.112 | 14.580 | <0.001 | 0.65(0.52–0.81) |
| Smoking status | | | | | |
| Non-smoker | – | – | – | – | 1.00 |
| Smoker | −1.360 | 0.136 | 1.011 | 0.315 | 0.87(0.70–1.14) |
| Alcohol consumption | | | | | |
| Non-drinker | – | – | – | – | 1.00 |
| Drinker | 0.173 | 0.131 | 1.742 | 0.187 | 1.19(0.92–1.54) |
| Shift work status | | | | | |
| No shift work | – | – | – | – | 1.00 |
| Shift work | 0.797 | 0.114 | 48.849 | <0.001 | 2.22(1.78–2.78) |

**Table 2** (*continued*)

| Item | b | SE | Wald χ² | P value | OR (95%CI) |
|---|---|---|---|---|---|
| BMI, kg m⁻²(Body mass index, kg m⁻²) | | | | | |
| 18.5∼24 | – | – | – | – | 1.00 |
| <18.5 | −0.018 | 0.620 | 0.001 | 0.977 | 0.98(0.29–3.31) |
| >24 | 0.883 | 0.118 | 55.753 | <0.001 | 2.42(1.92–3.05) |
| Length of service, years | | | | | |
| <10 | – | – | – | – | 1.00 |
| 10–20 | 0.849 | 0.250 | 11.541 | 0.001 | 2.34(1.43–3.81) |
| >20 | 1.354 | 0.243 | 30.941 | <0.001 | 3.87(2.40–6.24) |

**Table 3  Relationship between sleep duration, sleep quality and hypertension after stratification.**

| Item | Sleep duration, h | | | | | Sleep quality | | |
|---|---|---|---|---|---|---|---|---|
| | 7–8 | <7 | P | >8 | P | Good | Poor | P |
| Gender[a] | | | | | | | | |
| Female | 1.00 | 1.54(1.16–2.04) | 0.003 | 1.62(0.75–3.47) | 0.218 | 1.00 | 1.78(1.26–2.49) | 0.001 |
| Male | 1.00 | 1.49(1.00–2.23) | 0.049 | 1.63(0.58–4.58) | 0.350 | 1.00 | 1.47(0.88–2.46) | 0.142 |
| Age, years[b] | | | | | | | | |
| <30 | 1.00 | 9.03(2.32–35.15) | 0.002 | 3.89(0.27–42.08) | 0.342 | 1.00 | 5.24(0.35–78.90) | 0.232 |
| 30–45 | 1.00 | 1.59(1.14–2.20) | 0.006 | 0.79(0.23–2.68) | 0.707 | 1.00 | 1.71(1.10–2.66) | 0.017 |
| >45 | 1.00 | 1.27(0.91–1.77) | 0.161 | 3.36(1.42–7.91) | 0.006 | 1.00 | 1.60(1.09–2.34) | 0.016 |
| Shift work status[c] | | | | | | | | |
| Shift work | 1.00 | 1.55(1.16–2.07) | 0.003 | 0.98(0.39–2.46) | 0.967 | 1.00 | 1.70(1.17–2.47) | 0.005 |
| No shift work | 1.00 | 1.48(1.02–2.15) | 0.039 | 1.92(0.76–4.82) | 0.165 | 1.00 | 1.60(1.03–2.48) | 0.037 |
| Length of service, years[d] | | | | | | | | |
| <10 | 1.00 | 4.08(1.92–8.83) | <0.001 | 1.68(0.35–8.11) | 0.520 | 1.00 | 0.74(0.31–1.74) | 0.488 |
| 10–20 | 1.00 | 2.79(1.59–4.86) | <0.001 | 0.82(0.10–6.83) | 0.855 | 1.00 | 3.01(1.23–7.35) | 0.016 |
| >20 | 1.00 | 1.20(0.92–1.56) | 0.191 | 2.25(1.06–4.79) | 0.035 | 1.00 | 1.64(1.18–2.27) | 0.003 |

Notes.
[a]Adjust factors: age, ethnicity, income and shift work status, length of service.
[b]Adjust factors: gender, ethnicity, income and shift work status, length of service.
[c]Adjust factors: age, gender, ethnicity, income and length of service.
[d]Adjust factors: age, gender, ethnicity, income and shift work status.
The *OR* value and *95%* confidence interval are listed in the table.

a cardiovascular risk screening in 31 provinces across China, with a cumulative screening of more than 1.7 million urban and rural residents aged 35 to 75 years old. The results showed that the detection rate of age-adjusted hypertension was 37%. However, the prevalence of hypertension in Xinjiang oil workers was 15.3% in this study. This difference could be attributed to the following reasons: first of all, the prevalence of hypertension varies with the age composition of the population, increasing with a greater proportion of elderly people. In this study, the number of oil workers < 45 years old in this survey was relatively large, accounting for about 67% of the sample population. Secondly, Karamay City had adopted a comprehensive "four-party linkage" prevention and control model integrated with hospitals, community health service agencies, disease prevention and control agencies, and

residents in the management of chronic diseases. Consequently, remarkable results have been achieved, and it has already been recognized as the Chinese Model Region for Chronic Disease Management in 2019. In addition, the sleep status of specific occupational groups may result in different prevalence of hypertension depending on their working conditions and working environment (*Magnavita et al., 2019*). Nevertheless, the prevention and control of hypertension of oil workers still requires intervention.

Most oil-producing fields in Xinjiang are located in remote areas of the Gobi Desert, with harsh natural conditions. Oil field workers are not only affected by severe weather, but also are under pressure from long working hours together with an irregular shift system, resulting in lack of sleep (*Tao et al., 2015*). From a physiological basis under such conditions, dehydration and subsequent cardiovascular stress are more likely to occur (*Zhang et al., 2019*). While some studies have shown unclear association between sleep and hypertension, other studies have shown that sleep deprivation increases the sympathetic activity of the nervous system and changes the hypothalamus-pituitary-adrenal axis, resulting in increased cortisol levels, and elevation of blood pressure and heart rate, identifying poor sleep quality an important risk factor for hypertension (*Bruno et al., 2013*; *Feng et al., 2019*).

## Gender differences in the effect of sleep on hypertension

There is, however, a growing recognition of gender disparity in sleep-wake and circadian rhythm disorders (*Mong & Cusmano, 2016*; *Nishichi et al., 2013*). Analyzing gender by stratification in our study confirms an increased prevalence of poor sleep quality and insufficient sleep duration in females, compare to males. An explanation for the gender difference could be that men and women have different occupational trajectories and different social support in the workplace (*Hayase, Shimada & Seki, 2014*). Physiologically, hormonal changes in the female menstrual cycle, menopause, pregnancy, and postpartum affect the body's circadian rhythm and sleep architecture, leading to frequent sleep disturbances and worsening sleep quality. In addition, these hormonal changes are associated with depression, anxiety and irritability, which may additionally cause a deterioration in the quality of sleeping in females. In terms of psychosocial factors, work stress and family stress aggravate the development of sleep problems, including insomnia, and females are more likely to be vulnerable to effects caused by a stressful life, because of a greater share of household duties and taking care of children in addition to work, resulting in more sleep problems.

## Age differences in the effect of sleep on hypertension

Further, sleep disturbance is common during the menopausal transition and its effect of sleep duration on hypertension is U-shaped (*Kecklund & Axelsson, 2016*). We revealed similar findings in our study after stratifying for age. We found insufficient sleep duration is linked to hypertension in those < 45 years of age, while too long sleep duration leads to a similar result in those > 45 years, consistent with the results of *Kecklund & Axelsson (2016)*. A large number of studies have shown that insufficient sleep increases the excitability of the sympathetic nervous system and the renin-angiotensin system. This may be related

to increased catecholamine synthesis in the central nervous system, which young people are more sensitive to. On the other hand, the sleep duration of the elderly is affected by age-related sleep structure changes, compensatory daytime sleep, and drug side effects, resulting in an increase in the prevalence of hypertension (*Lu et al., 2015*).

## Shift work differences in the effect of sleep on hypertension

In addition, it is well-recognized that long-term shifts of may have some adverse health outcomes (*Manohar et al., 2017*). Therefore, we analyzed whether there is a difference in the effect of shift work on the relationship between sleep and hypertension. We observed slightly increased association between sleep disturbances and extension of shift work. Studies have confirmed that shift work is associated with disruption of sleep patterns and circadian rhythms. In particular, night shifts can disturb chronobiological rhythms and reduce the secretion of melatonin, thus directly reducing the quality of sleep (*Guo et al., 2017*). Shift work also interferes with the quality of sleep, which can become chronic and remain even after exposure has ceased. It has been reported that even after retirement older workers who worked shifts have a worse sleep pattern than other retirees (*Härmä et al., 2018*). In these former workers, polysomnographic studies have demonstrated the existence of a direct relationship between the duration of shift work and the frequency of altered sleep patterns (*Heath et al., 2016*).

## Length of service differences in the effect of sleep on hypertension

Finally, after stratification according to the length of service, we found decrease in length of service increases the risk of hypertension when sleep duration is insufficient. But poor sleep quality is only associated with oil workers who have worked > 10 years. This is likely because the length of service is highly related to age, and most of those with shorter length of service are young and middle-aged (*Elogue et al., 2014*).

Based on the above findings, to prevent occurrence of hypertension, we recommend strengthening health education for oil workers, including the provision of guidelines to help them achieve an appropriate work-life balance to maintain their health and adopt good sleeping habits such as changing sleep patterns. At the same time, employers should improve the working environment, and establish an appropriate system to improve the quality of professional life for workers, taking into consideration that shift work has an adverse impact on health. Finally, considering hypertension patients, health promotion and education should be strengthened to help them achieve the best therapeutic effect, thereby improving their long-term quality of life and health.

Although this study has some important findings, several limitations should be acknowledged. First, the participants were oil workers working in desert areas, who may be prone to occupational stress due to their specific occupational particularities. *Magnavita & Garbarino (2017)* have suggested occupational stress is an important factor associated with sleep disorders. On the other hand, some studies have indicated that sleep is an important moderator of the relationship between stress and hypertension (*Garbarino & Magnavita, 2019*). Therefore, owing to the complex relationship between sleep and hypertension, we did not include the impact of occupational stress in this initial study. In a future study,

we plan to explore the relationship among sleep, occupational stress, and hypertension through intermediary effect analysis. Second, sleep quality and duration are self-reported data. There were no standard cut-off values to judge normal or abnormal sleep duration, or good or poor sleep quality. Third, causality and temporality could not be ascertained because this study was a cross-sectional survey. And finally, because of the sample size, it cannot represent all the oil workers in China. Thus, further investigation of oil workers is warranted for further understanding of the relationship between sleep and hypertension.

## CONCLUSIONS

Our findings suggest that sleep disturbances are associated with the prevalence of hypertension among oil workers. Our study confirms that insufficient sleep duration and poor sleep quality are factors associated with hypertension. After stratification by gender, age, shift work status, and length of service, when both insufficient sleep duration and poor sleep quality coexist, the following were found: there is an increased prevalence of hypertension in females; the risk of hypertension decreases with age and length of service; And the prevalence of hypertension in shift workers is high. This study has identified a number of factors associated with hypertension in oil workers doing shift work in relation to sleep and hypertension, providing valuable data that can be used to draw up measures and guidelines to prevent and manage hypertension among this and other occupational groups, especially those with shift work, and hypertension patients in general.

## ACKNOWLEDGEMENTS

The authors thank all participants and investigators.

### Funding

This study was supported by public health and preventive medicine, a key discipline of the 13th five-year plan in the Xinjiang Uygur Autonomous Region (No. 9911091113404), the 64th Western region postdoctoral talent subsidy program (No. 2018M643826XB), and the Department of Education, Xinjiang Uygur Autonomous Region (No. XJEDU2018Y029). The funders had no role in study design, data collection and analysis, decision to publish, or preparation of the manuscript.

### Grant Disclosures

The following grant information was disclosed by the authors:
Xinjiang Uygur Autonomous Region: 9911091113404.
64th Western Region Postdoctoral Talent Subsidy Program: 2018M643826XB.
Department of Education, Xinjiang Uygur Autonomous Region: XJEDU2018Y029.

### Competing Interests

The authors declare there are no competing interests.

## Author Contributions

- Fen Yang conceived and designed the experiments, performed the experiments, analyzed the data, prepared figures and/or tables, authored or reviewed drafts of the paper, and approved the final draft.
- Yuanyue Zhang and Ruiying Qiu performed the experiments, prepared figures and/or tables, and approved the final draft.
- Ning Tao conceived and designed the experiments, analyzed the data, authored or reviewed drafts of the paper, and approved the final draft.

## Human Ethics

The following information was supplied relating to ethical approvals (i.e., approving body and any reference numbers):

The Ethics Committee of the First Affiliated Hospital of Xinjiang Medical University granted approval to carry out the study within its facilities (Ethical Application Ref: 2015006).

## Data Availability

The raw measurements are available in the Supplementary Files.

## Supplemental Information

Supplemental information for this article can be found online at http://dx.doi.org/10.7717/peerj.11318#supplemental-information.

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
