# Peer review of "Association of sleep duration and sleep quality with hypertension in oil workers in Xinjiang"

_PeerJ, doi:10.7717/peerj.11318_

## Round 0.1 · original submission · Major Revisions

Comments by reviewers must be addressed.

·

Basic reporting

No comment

Experimental design

The study investigates the relationship between sleep problems and hypertension.

In the Introduction, among the numerous factors associated with hypertension (L.41), the authors forget to include stress.

In the review by Bergmann et al. [The appraisal of chronic stress and the development of the metabolic syndrome: a systematic review of prospective cohort studieshttp://www.endocrineconnections.org DOI: 10.1530/EC-14-0031] many studies on the relationship between chronic stress and hypertension are reported.

Work stress is significantly associated with sleep problems.

Sleep problems, in fact, increase the sensibility to work stress [Magnavita N, Garbarino S. Sleep, Health and Wellness at Work: A Scoping Review. Int J Environ Res Public Health. 2017 Nov 6;14(11). pii: E1347. doi: 10.3390/ijerph14111347[, and stress increases sleep problems.

Sleep acts as a moderating factor in the relationship between stress and hypertension [Garbarino S, Magnavita N. Sleep problems are a strong predictor of stress-related metabolic changes in police officers. A prospective study. PLoS One. 2019 Oct 22;14(10):e0224259. doi: 10.1371/journal.pone.0224259].

This clarification is necessary to clarify the complexity of the factors influencing hypertension.

Unfortunately, the study did not take into account occupational stress, which can be important for hypertension and also for sleep problems.

This is a limitation of the study, which should be discussed by the authors.

Validity of the findings

The major limitation of the study is that it is a cross-sectional study. It is not legitimate to infer about causality in this type of study. In fact, there could be a reverse causation, the fact of suffering from hypertension could disturb sleep. Authors should discuss this point. They should avoid the causal interpretation of their findings, and change their claims non-purposely. It will be fair to say that impaired sleep is associated with hypertension, not that it causes hypertension.

Additional comments

No comment

Reviewer 2 ·

Basic reporting

The author mentioned that the desert petroleum workers are a special group of people, the author should give more instruction of the population. Many similar studies has been determined previous and this study in some extent is lack of innovation.

Experimental design

Methodology section:
1) Many important variables should given more instruction, such as the definition of smoking, drinking and shift work
2) Lack of some important references, for example, PSQI questionnaire.
“The Pittsburgh Sleep Quality Index (PSQI) was used to assess participants' subjective sleep quality and sleep duration. The survey used the Chinese version of the Pittsburgh Sleep Quality Index (CPSQI), which was Cronbach's alpha of 0.89 and a KMO of 0.91. CPSQI consists of 7 factors and there are 19 items. Each item is scored from 0 to 3 according to the 4-level Likert scale. In this study, CPSQI ≤ 5 was considered ad poor sleep quality, whereas a score of >5 was regarded as poor sleep quality. Sleep duration was classified into “insufficient” (﹤7 h), “normal”(7- 8h), and “long”(>8 h) based on previous reports.”

Validity of the findings

Result section: Since this study is a cross-sectional study, it cannot be determined causal relationship. It not correct that poor sleep quality are risk factors for hypertension.

Discussion part: The content of the discussion is not clear and lack of logic.
1)The author has missed some discussion of the relationship between some variables and hypertension.
2) In addition the result of this study suggested the prevalence of hypertension was 15.3%, which was significant lower than the figure that 37% published in previous study (Jiang Lixin et al., 2017). The author should discuss the difference in the rate of hypertension.

Additional comments

The content of this study and the languages should be improved.

Reviewer 3 ·

Basic reporting

The paper is clear, but english could be improved.

Experimental design

1)Many important variables should given more instruction, such as the definition of smoking, drinking and shift work
2) Lack of some important references, for example, PSQI questionnaire.
“The Pittsburgh Sleep Quality Index (PSQI) was used to assess participants' subjective sleep quality and sleep duration. The survey used the Chinese version of the Pittsburgh Sleep Quality Index (CPSQI), which was Cronbach's alpha of 0.89 and a KMO of 0.91. CPSQI consists of 7 factors and there are 19 items. Each item is scored from 0 to 3 according to the 4-level Likert scale. In this study, CPSQI ≤ 5 was considered ad poor sleep quality, whereas a score of >5 was regarded as poor sleep quality. Sleep duration was classified into “insufficient” (﹤7 h), “normal”(7- 8h), and “long”(>8 h) based on previous reports.”

Validity of the findings

Result section: Since this study is a cross-sectional study, it cannot be determined causal relationship. It not correct that poor sleep quality are risk factors for hypertension.

Discussion part: The content of the discussion is not clear and lack of logic.
1)The author has missed some discussion of the relationship between some variables and hypertension.
2) In addition the result of this study suggested the prevalence of hypertension was 15.3%, which was significant lower than the figure that 37% published in previous study (Jiang Lixin et al., 2017). The author should discuss the difference in the rate of hypertension (cite also: Magnavita N, Di Stasio E, Capitanelli I, Lops EA, Chirico F, Garbarino S. Sleep Problems and Workplace Violence: A Systematic Review and Meta-Analysis. Front Neurosci. 2019;13:997. Published 2019 Oct 1. doi:10.3389/fnins.2019.00997)

Additional comments

The paper is good, but you should improve some aspects of the paper.

---

## Round 0.2 · Minor Revisions

Comments on 50421

In addition to reviewer #1's comments, take into consideration these points:

Overall- suggest copyediting

Abstract: It would be useful to add a sentence at the beginning outlining why this study is useful or important. Please add “our results show” at the start of the concluding sentence.

Methods:

1. Please provide a copy of the questionnaire used in the study, in the original language and an English translation.
2. Please clarify what you mean by “unfinished blood pressure tests”. At this point you have not mentioned who was doing the tests for blood pressure and why, so this point is confusing. Please revise.
3. Please clarify what you mean by “severe organic diseases”
4. Please also mention whether the measurement of blood pressure was carried out three times at the same time or during different times of the day.
5. Statistical methods:
a. I would like to see the sample size calculation for this study. It is important to know if you have enough subjects so as to answer your research question.
b. Pearson or Fisher Chi squared tests?
c. “logistic regressions were performed to evaluate the relationship of sleep duration and sleep quality with hypertension”. More details about this should be given. Which predictors were included in the model? Did you calculate OR? How did you assess the goodness-of-fit of the models? Discrimination and calibration? It seems to me that you have used a standard phrase to describe this part of the text.
d. “the test level was 0.01 according to the large sample”. You have 3,040 subjects which is not a great number. Please, use the standard type I error (5%), and consequently update your results and discussion, and methods if required, because I do not know how you selected the predictors and this could update the models.
Results
1. Please provide exact p-values throughout, unless p<0.001. Please also provide 95% confidence intervals along with all p-values

Discussion
1. Please summarise your study in the first few sentences of the discussion, including your aims, methodology and findings. For example, as you have done at line 187 “In this study, we adjusted for gender, age, and other confounding factors, and performed logistic 188 regression analysis to assess the relationship between sleep duration and sleep quality, and 189 hypertension. We found that insufficient sleep duration and poor sleep quality are associated with 190 hypertension in Xinjiang oil workers”. For clarity, please move these sentences to the start of the discussion section.
2. Please organise the discussion as follows: brief summary of the study and your findings, followed by brief comparison to relevant literature, strengths and limitations of the study concluding with implications for policy or future research. Please add sub-headings for readability.
3. Please remove the sentence “154 Social pressure on the professional population increases enormously with gradual acceleration of 155 the pace of life (Zhang et al., 2014)” as you did not test social pressure in your population.
4. Please remove lines 172-186 as this is largely known about risk factors of hypertension
5. I suggest lines 200-217 are removed as many of these claims cannot be substantiated by scientific evidence. Following lines 198-199 where you mention sleep quality as a risk factor for hypertension, please add 1-2 sentences on sleep quality in women and how your findings confirm existing evidence that women may be more vulnerable to poor sleep quality.
6. Please remove lines 233-245 as they are not relevant
7. Limitations: you may include a sentence or two about whether the population studied in this submission is representative of all oil workers in China, for example.

Conclusions
Line 273- I suggest removing “In sum” and instead starting with “our findings suggest” or “our results show” or similar.

·

Basic reporting

The authors have modified the text, introducing new references. However, their work needs careful review, because some items are cited twice, and other references are inappropriate. For example, at line 261 the appropriate references are (Magnavita N, Garbarino S. Sleep, Health and Wellness at Work: A Scoping Review. Int J Environ Res Public Health. 2017;14(11). pii: E1347. doi: 10.3390/ijerph14111347), and (Garbarino S, Magnavita N. Work stress and metabolic syndrome in police officers. A prospective study. PLoSOne, 2015; 10 (12):e0144318. doi: 10.1371/journal.pone.0144318).

The sentence reported on line 263 "some studies have shown an ambiguous relationship among sleep, occupational stress, and hypertension" is inappropriate. The study cited [Garbarino S, Magnavita N. Sleep problems are a strong predictor of stress-related metabolic changes in police officers. A prospective study. PLoS One. 2019;14(10): e0224259. doi: 10.1371/journal.pone.0224259] indicates that sleep is an important moderator of the relationship between stress and hypertension or other components of the metabolic syndrome.

Experimental design

Research question well defined, relevant & meaningful.

Validity of the findings

Data are robust, statistically sound, & controlled.

Additional comments

The authors have modified the text, introducing new references. However, their work needs careful review, because some items are cited twice, and other references are inappropriate. For example, at line 261 the appropriate references are (Magnavita N, Garbarino S. Sleep, Health and Wellness at Work: A Scoping Review. Int J Environ Res Public Health. 2017;14(11). pii: E1347. doi: 10.3390/ijerph14111347), and (Garbarino S, Magnavita N. Work stress and metabolic syndrome in police officers. A prospective study. PLoSOne, 2015; 10 (12):e0144318. doi: 10.1371/journal.pone.0144318).

The sentence reported on line 263 "some studies have shown an ambiguous relationship among sleep, occupational stress, and hypertension" is inappropriate. The study cited [Garbarino S, Magnavita N. Sleep problems are a strong predictor of stress-related metabolic changes in police officers. A prospective study. PLoS One. 2019;14(10): e0224259. doi: 10.1371/journal.pone.0224259] indicates that sleep is an important moderator of the relationship between stress and hypertension or other components of the metabolic syndrome.

---

## Round 0.3 · accepted · Accept

All the reviewers' concerns have been correctly addressed.